# Teacher experiences and understanding of citizen science in Australian classrooms

**Larissa Braz Sousa**[1,2]*, **Ciara Kenneally**[1], **Yaela Golumbic**[1,3], **John M. Martin**[4,5], **Christine Preston**[6], **Peter Rutledge**[1,2], **Alice Motion**[1,2]*

1 SCOPE Group, School of Chemistry, The University of Sydney, Camperdown, New South Wales, Australia, 2 Charles Perkins Centre Citizen Science Node, The University of Sydney, Camperdown, New South Wales, Australia, 3 The Steinhardt Museum of Natural History, Tel Aviv University, Tel Aviv, Israel, 4 Institute of Science and Learning, Taronga Conservation Society Australia, Mosman, New South Wales, Australia, 5 School of Life and Environmental Sciences, The University of Sydney, Camperdown, New South Wales, Australia, 6 Sydney School of Education and Social Work, The University of Sydney, Camperdown, New South Wales, Australia

* larissa.brazsousa@sydney.edu.au (LBS); alice.motion@sydney.edu.au (AM)

**Data Availability Statement:** All relevant data are within the paper and its Supporting information files. Some columns and rows in Supplementary

## Abstract

Citizen science represents an important opportunity for school students to make real-world connections with science through context-based learning with the potential to increase their engagement, enjoyment and understanding of science. However, to date, citizen science has not experienced wide uptake in school settings and there is a paucity of information about its implementation in the classroom. Here we present a mixed-method approach investigating teachers' knowledge and use of citizen science in Australian classrooms. We explored teachers' experience and perceptions of citizen science, and opportunities and barriers to incorporate citizen science as an educational approach through an online questionnaire. Among the teachers surveyed, 45% (n = 295) had personally participated in citizen science outside of school and 41% (n = 283) had incorporated citizen science projects in classroom lessons. Teachers (45%, n = 295) reported participating in citizen science initiatives multiple times. Also, most projects that teachers were involved in (77%, n = 292) were related to ecological studies, such as species monitoring. Citizen science was reported to be a relatively new approach; used by teachers for less than a year on average. The main challenges included a lack of knowledge, time, confidence, and clarity regarding citizen science project alignment with the Australian curriculum. Additionally, 92% of respondents said they would be more encouraged to use citizen science in classrooms if projects were aligned to the curriculum. Identifying ways to increase teachers' openness to incorporate citizen science in their classrooms is crucial to its successful widespread, long-term, and meaningful implementation. Encouraging broader participation of teachers in citizen science based on their previous experiences could address their expectations and increase their confidence and feeling of ownership. These research findings suggest that meaningful and applicable citizen science programs could be co-created by addressing resource limitations and curriculum alignment challenges. Implementing solutions to these barriers is likely to contribute to the development of sustainable school-inclusive citizen science projects, with potential to positively impact science education in the long-term.

Material 2 were anonymised to comply with the Ethics requirements.

**Funding:** This research was funded by the NSW Department of Education via a Strategic Education Grant (AUD600,000) awarded to AM, YG, CP, AB, and JM with additional support of Motion through a Westpac Research Fellowship. The funders did not play any role in the study design, data collection and analysis, decision to publish or preparation of the manuscript.

**Competing interests:** The authors have declared that no competing interests exist.

## Introduction

Citizen science encompasses a range of diverse approaches, with applications across many disciplines and contributions by participants from different countries and of all ages, that has been defined as public participation in scientific research [1–3]. In the past two decades alone, thousands of citizen science projects have engaged at least 6 million participants globally [1,4]. These projects have produced hundreds of publications [1,5–7] and countless opportunities for education [8–10], connection with nature [11], and community interaction [9,12]. Citizen science has been explored as a way to improve educational outcomes and contribute to scientific knowledge development [13–16].

Educational outcomes are increasingly an aim of citizen science, often with a focus on science education for young people [17,18]. Citizen science projects may prioritise education as an outcome of participation [19]. Although the benefits of school-based citizen science have been explored, limited studies describe citizen science projects designed for schools and assess their benefits for students, teachers, and the entire community [20–22]. While some studies report citizen science together with an integrative approach linking science education and environmental education [23–25] there is a clear gap in research into citizen science integration into other subjects and disciplines [20,26]. Of the education-based citizen science projects reported in the literature most are contributory, with fewer examples of co-designed projects [27,28].

There are, however, limited examples of citizen science projects that have been specifically designed to integrate pedagogical objectives and approaches, aiming to enhance teaching practices and learning outcomes within educational settings [29,30]. Examples include programs that provide structured teaching resources and adaptable lessons to support schools' pedagogical practices and curricula [30,31]. Such programs have been especially reported in the fields of environmental science and biodiversity [32,33].

There are several documented benefits for students who participate in citizen science, including physical, emotional, interpersonal, social, and educational [18]. For example, students were found to be more motivated to attend school and enjoyed their classes more after participating in citizen science [34]. This increase in interest and enjoyment at school was mirrored in student attitudes towards science, with citizen science participation maintaining or increasing students' positive science outlook [35]. Benefits for students were attributed to the core nature of citizen science; where students are active learners involved in scientific research. Furthermore, research has shown that students benefit from contributing to citizen science as they advance their learning of scientific concepts and the scientific enquiry process [18]. This learning can enhance students' understanding of real world science as they make first-hand contributions to research [21]. Citizen science projects can also provide students with opportunities to develop knowledge and skills in areas outside of Science, Technology, Engineering, and Mathematics (STEM), such as social sciences and humanities [34,36–39].

Researchers have explored the impact of citizen science in making scientific knowledge more accessible [40,41]. To do that, citizen science needs to consider approaches that are more empowering and that embrace participants roles beyond data collection to include aspects such as research design, discussions, and implications [42,43]. In order to achieve the best outcomes, citizen science projects collaborating with, or being implemented by schools need to be cognisant of the particular abilities and educational needs of participating teachers and students [13]. The framing of research tasks is particularly important to match students' skills in this context. For example, a citizen science project in rural northern Germany engaged 8- to 10-year-old students to investigate seed predation. While students struggled to estimate vegetation height, the values they recorded for seed counting were similar to those of professional

scientists, as their skills were matched appropriately to the methods [44]. Guidance surrounding the appropriate skill and knowledge level of students is more readily achieved where citizen science projects are curriculum aligned [45]. Teachers have reported that projects developed without clear links to the curriculum are challenging to incorporate into lessons [18,29,46], this may be due to teacher time constraints and/or because project tasks are mismatched with student skill level. Ensuring that student skill level is both considered and understood by the citizen science project leaders, ideally through a co-design process with teachers, is essential for the optimal success of citizen science projects in engaging schools and students [47].

In addition to the learning benefits for students engaging in citizen science, teachers, scientists, and projects also benefit from these partnerships. For instance, teachers are able to work closely with researchers and citizen science coordinators [48]. This may increase teachers' confidence in supervising and facilitating citizen science projects and, more generally, in teaching science, which may be advantageous for teachers who have not been scientifically trained at the tertiary level [49,50]. Scientists also benefit as they are able to expand the amount, depth or scope of data collected and increase the reach and impact of their research through student engagement in citizen science [29]. Furthermore, while citizen science participation is skewed to white, middle-aged men with higher education levels and a large interest in science [51,52], engagement of school communities can reduce demographic and geographic bias in data collection and increase inclusiveness in citizen science by engaging commonly underrepresented groups [53]. For example, school participation can increase cultural diversity or broaden participants' age-range [29].

Although citizen science can bring several benefits to schools and communities, there are still specific barriers to its local and global implementation [30,34,45,54,55]. Analysing teachers' experiences and needs is crucial to understand the challenges and opportunities for effective school-based citizen science implementation.

This article describes work and key outcomes towards Stage One of the Learning By Doing (LBD) initiative. LBD is a multidisciplinary research project funded by the New South Wales (NSW) government Department of Education. It is a joint initiative of an interdisciplinary group of researchers committed to exploring the multiple learning dimensions of citizen science projects. The program was launched in 2021 and comprises three phases, 1) exploring the breadth and depth of citizen science in Australian schools, 2) evaluating the learning dimensions of citizen science, and 3) mapping citizen science to the Australian curriculum [56].

This study aimed to examine the current use of citizen science in Australian schools, together with the benefits and barriers experienced by teachers specialising in different subjects towards the wider implementation of citizen science. We investigated four focal research questions: 1) What knowledge and experience do teachers have of citizen science? 2) How have teachers used citizen science in their personal life and with their students? 3) What are the barriers to citizen science implementation in schools and its broader use? and 4) What would encourage and support the future participation of teachers in citizen science? By addressing these research questions, we aimed to identify future directions that support the implementation of school-based citizen science with the broader aim of enhancing science acceptance, science participation, and scientific literacy skills among school students.

## Methods

This study used a mixed-method approach to enable complementary viewpoints addressing the same research questions [57,58]. An online questionnaire comprising open-ended and close-ended questions was designed for distribution among teachers across Australia (S1 File). The questionnaire was hosted on the online cloud-based questionnaire platform Qualtrics

[59], available through The University of Sydney. Data was collected in Qualtrics and exported into Microsoft Excel and R [60] for analysis. Data collection spanned five months (10[th] August 2021—31[st] December 2021).

## Questionnaire design

The first section of the questionnaire focused on participant demographics and teaching experience. Information about each participant's school, including the postcode and type of school (student age and educational stage, private/independent, and co-ed/single sex) was collected.

The second section focused on understanding respondents' knowledge of citizen science outside the school environment. Respondents were asked whether they had heard of citizen science and to then select a definition of citizen science from a sample of possible answers. Following this, participants were asked about their use of citizen science in a personal context. If participants had used citizen science they were asked for details of their experience, including the projects they had participated in and how frequently. If participants had not previously engaged in citizen science, they were asked to explain why by selecting the answers that applied from a predefined list (e.g., not aware of citizen science; not interested). When questioning teachers regarding their knowledge of citizen science, the authors assume teachers to have varying levels of familiarity with the concept and practice of citizen science and that teachers' understanding of citizen science may influence their willingness and ability to incorporate such projects into their teaching practices.

The third section targeted citizen science use within schools. Teachers were asked if they had used citizen science in their lessons, with the questionnaire designed to branch according to their response. They were also asked about the projects they had been involved with, the school stages, and the extent of their involvement. Finally, they were asked whether they received support from research teams to participate in citizen science.

Participants who had not used citizen science within the school environment were asked what had limited their involvement. Both groups of teachers were asked to detail what interventions or resources would support their future involvement in citizen science and the factors that would make them more likely to participate.

## Research tools validation

Content validity was assessed by expert judges [61], composed of six current or former teachers based at Taronga Zoo, connected to one of the authors [JMM]. All of these teachers had been involved in citizen science in their classrooms, regardless of teaching background. The teacher group completed the questionnaire and provided feedback regarding clarity and order of the questions. Multiple-choice options were revised and extended following their feedback, in order to better represent teacher experience.

## Questionnaire distribution and collection

Following validation, the questionnaire was distributed by email to personal networks by the authors and linked to posts on Twitter, Facebook, and Instagram. These emails and social media posts also encouraged participants to share the questionnaire within their networks—snowball sampling technique [62]. Additionally, organisations known to researchers were contacted to further disseminate the questionnaire within their teacher networks. Teacher specific Facebook pages were also used to further disseminate the questionnaire.

## Data analysis

Quantitative data analysis included descriptive statistics to quantify the number of responses, and chi-square tests to make comparisons between categorical variables [63]. These analyses were conducted using Microsoft® Excel® 2019 MSO (16.0.10366.20016) and R [60] after importing the data collected from Qualtrics (S3 File). The quantitative analysis in this questionnaire used descriptive statistics, representing the categorical data through proportions of respondents. From this, further analysis was conducted comparing variables. In all cases, chi-square tests were used, as comparisons were always conducted between two categorical variables.

Qualitative data collected from open-ended questions were exported and analysed to identify common themes. In this analysis, two authors [LBS and YG] generated initial codes, then searched, reviewed, and defined themes based on patterns within open-ended questions [64].

This research was approved by The University of Sydney's Human Research Ethics Committee [protocol number 2021/204]. The consent form was presented to respondents at the beginning of the questionnaire. Informed consent was obtained by participants completing and returning the questionnaire.

## Sample

A total of 355 teachers responded to the questionnaire, with 82% (n = 293) completing the full questionnaire. Respondents were predominantly women and relatively equally spread across four of five age ranges (Table 1). Respondents were predominantly secondary teachers (students aged 11–18 years). Most teachers surveyed were from co-ed schools (Fig 1).

Teachers had a range of experience levels, with 20+ years the most frequently selected category. A variety of science-specific tertiary experience was reported, with most teachers having no tertiary science experience and the second largest response corresponding to a third-year undergraduate level of science-specific study. Teachers reported that they taught an average of 2.5 subjects each (Fig 2). When asked about which subjects they teach (n = 297), 33% of teachers taught one subject and 30% taught four or more subjects. Of these, 58% of respondents teaching one subject were secondary school teachers (n = 98), and 74% (n = 88) of respondents teaching four or more subjects were primary school teachers. The most frequent combination of subjects were Science, Mathematics, English, Human Society and Its Environment (HSIE), Personal Development, Health and Physical Education (PDHPE), and Creative Arts for primary school teachers (28%, n = 157), and Science and Biology for secondary school teachers (12%, n = 182).

The school system in Australia is organised in stages or years. Primary School, from Kindergarten to Year 6 (K-6) comprises four stages of learning: 1) Early Stage or Foundation = Kindergarten (starting from 4.5 years of age); 2) Stage 1 = Years 1 and 2; 3) Stage 2 = Years 3 and 4; and 4) Stage 3 = Years 5 and 6. Secondary School, from Year 7 to Year 12 (Higher School Certificate) comprises three stages of learning: 1) Stage 4 = Years 7 and 8 (starting around 12 years of age); 2) Stage 5 = Years 9 and 10 (from about 14 years of age); and 3) Stage 6 = Years 11 and 12 (finishing ~18 years of age) [65].

Following the classification of postcodes, 77% of respondents were found to work in NSW, with smaller responses from teachers based in Victoria (8%), Queensland (6%), South Australia (3%), Northern Territory (3%), Western Australia (2%), Tasmania (1%), and Australian Capital Territory (1%). Due to the small sample sizes of some territories and states, statistically relevant comparisons between states/territories were not possible. Instead, postcodes were classified using the Department of Agriculture, Fisheries and Forestry classifications system [66]. The department has three postcode classifications: metropolitan, rural, and split (part

**Table 1. Teacher demographics from the online questionnaire.**

| Demographics | Responses | N | % |
|---|---|---|---|
| Gender | Female | 234 | 75% |
| | Male | 75 | 24% |
| | Prefer to self-describe | 1 | 0% |
| | Prefer not to say | 4 | 1% |
| Age | 20–29 | 59 | 19% |
| | 30–39 | 92 | 29% |
| | 40–49 | 76 | 24% |
| | 50–59 | 66 | 21% |
| | 60+ | 21 | 7% |
| Nationality | Australia | 271 | 86% |
| | Overseas | 43 | 14% |
| Primary language | English | 287 | 91% |
| | Other | 27 | 9% |
| Teaching experience (in years) | 1–5 | 78 | 25% |
| | 6–10 | 63 | 20% |
| | 11–15 | 54 | 17% |
| | 16–20 | 30 | 9% |
| | 20+ | 92 | 29% |
| Science background | No tertiary experience | 85 | 27% |
| | First year | 33 | 11% |
| | Second year | 21 | 7% |
| | Third year | 70 | 22% |
| | Honours | 38 | 12% |
| | Masters | 61 | 19% |
| | PhD or above | 6 | 2% |

metropolitan, part rural). We converted postcode entries on the survey using the postcode delivery classifications tool [66]. Out of 283 responses, 81% of teachers were from metropolitan areas, 16% from rural areas and 3% from split areas.

## Results

### Teachers' knowledge and experience about citizen science

Overall, over 40% of teachers had used citizen science in their personal lives (45%, n = 295) or lessons (41%, n = 283), and 83% of those who had not yet used it (n = 166) indicated that they would like to in their response. Teachers who participated as citizen scientists outside of school were significantly more likely to include it in their classrooms ($X^2$ = 51.36, df = 2, p < 0.05). Of teachers who had used citizen science in their classrooms (n = 108), they rated the experience mostly as average (35%), good (42%), and very good (13%). They reported using citizen science on multiple occasions (45%). Additionally, 94% of teachers who had used citizen science in their lessons intended to continue to do so.

Over half of respondents (51%, n = 298) had heard of 'citizen science'. Among these, 32% reported that their knowledge of citizen science was average, 33% reported a good knowledge, and 18% reported having a very good knowledge. When asked to select the definition that best describes citizen science, 53% (n = 298) selected the most correct from four options (see S2 File). Selecting the correct definition was correlated to school type (Table 2); teachers from

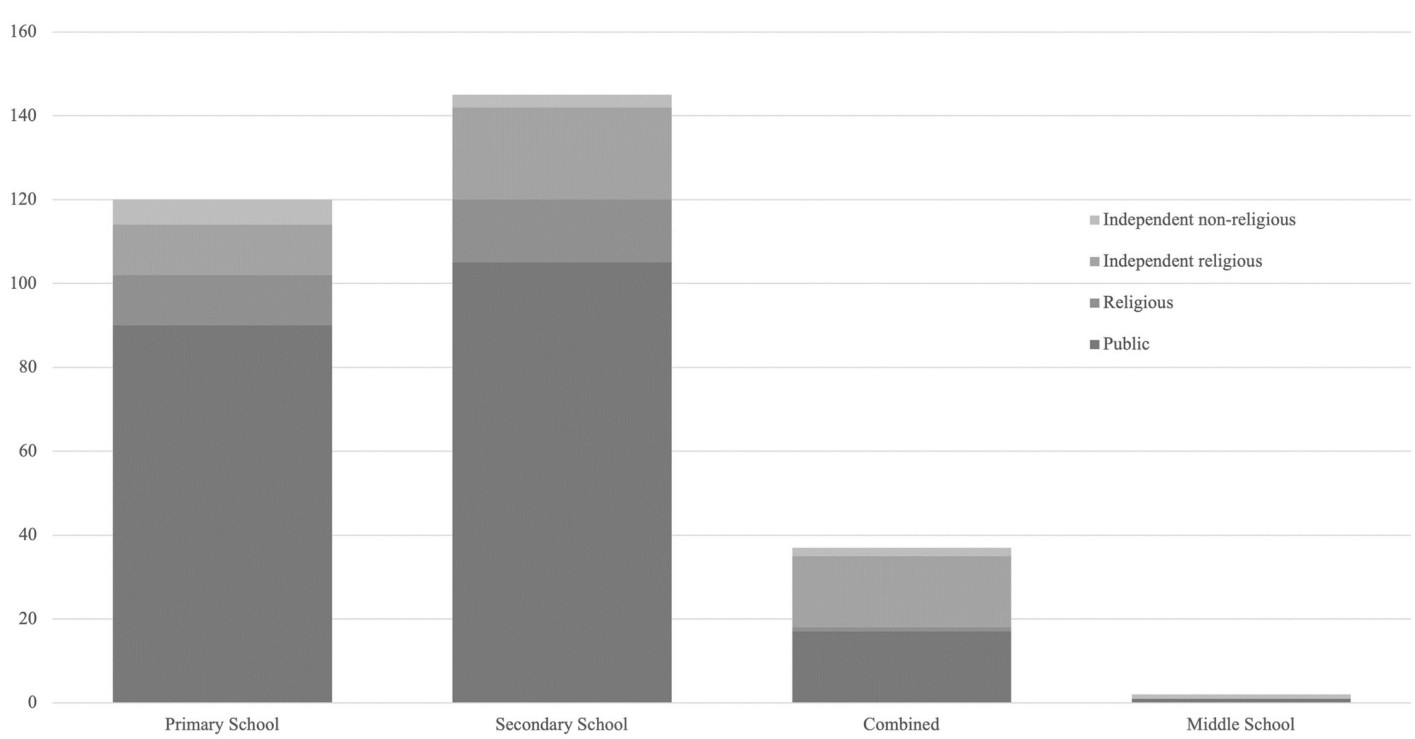

**Fig 1. School type and category of participating teachers (*N* = 288).**

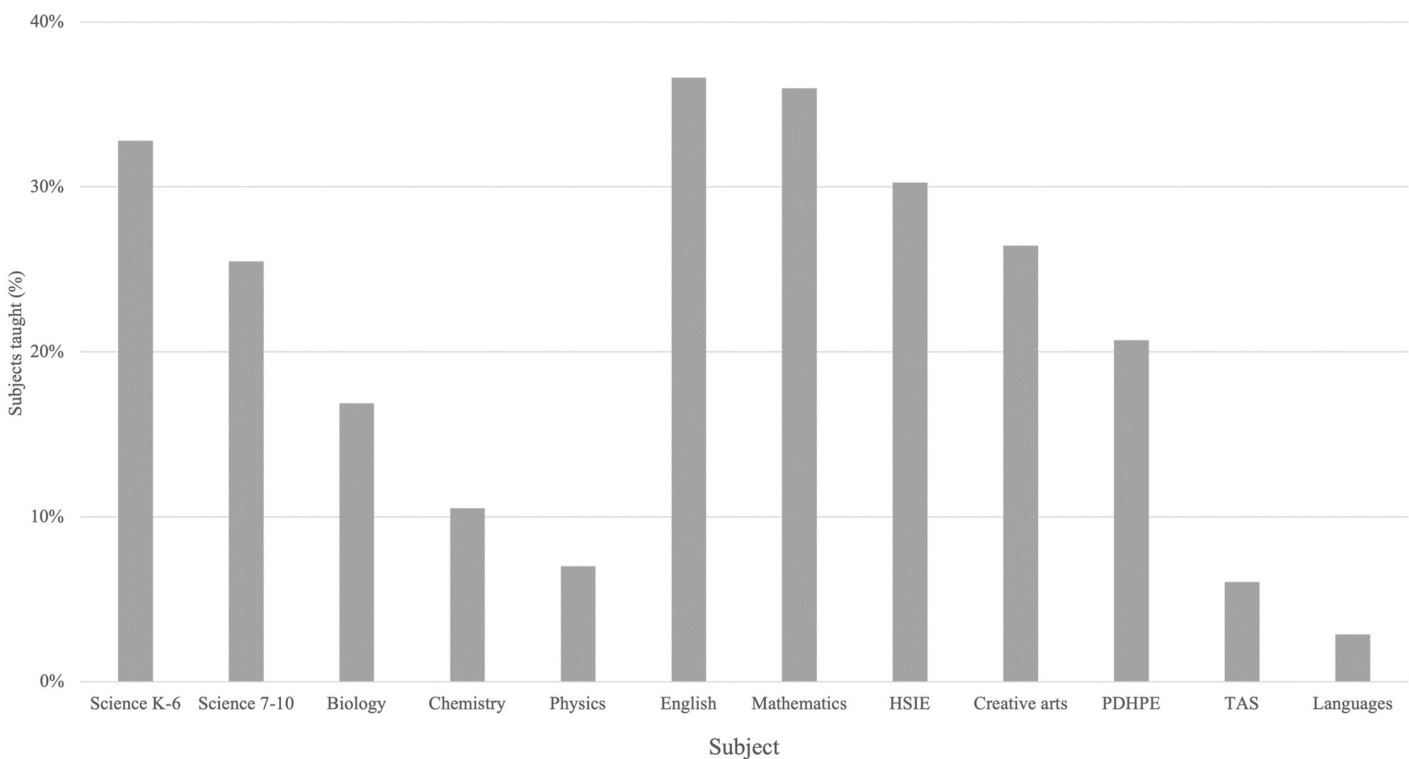

**Fig 2. School subjects taught by respondents (*N* = 297; teachers were able to select more than one subject).** Subjects are based on the subject areas for NSW. HSIE = Human Society and Its Environment; PDHPE = Personal Development, Health and Physical Education; TAS = Technological and Applied Studies.

**Table 2. Teachers' knowledge, experience, and use of citizen science in classrooms.**

| Cross-analysis between subject taught and school type and knowledge/experience with citizen science | | $X^2$ | df | p-value |
|---|---|---|---|---|
| Subject taught | Have heard of citizen science | 230.5 | 172 | <**0.05**\* |
| | Rate their own knowledge of citizen science (very poor, poor, average, good, very good) | 451.29 | 430 | 0.23 |
| | Correctly defined citizen science | 359.71 | 344 | 0.27 |
| | Have used citizen science in their classrooms | 216.49 | 172 | <**0.05**\* |
| School type (primary, middle, secondary, combined/central) | Have heard of citizen science | 19.36 | 6 | <**0.05**\* |
| | Rate their own knowledge of citizen science (very poor, poor, average, good, very good) | 31.52 | 15 | <**0.05**\* |
| | Correctly defined citizen science | 21.94 | 12 | <**0.05**\* |
| | Have used citizen science in their classrooms | 3.51 | 6 | 0.74 |
| School type (independent/non-religious, independent/religious, public, religious) | Have heard of citizen science | 17.13 | 6 | <**0.05**\* |
| | Rate their own knowledge of citizen science (very poor, poor, average, good, very good) | 32.88 | 15 | <**0.05**\* |
| | Correctly defined citizen science | 23.61 | 12 | <**0.05**\* |
| | Have used citizen science in their classrooms | 5.92 | 6 | 0.43 |
| School type (single sex boys, single sex girls, co-ed) | Have heard of citizen science | 11.42 | 4 | <**0.05**\* |
| | Rate their own knowledge of citizen science (very poor, poor, average, good, very good) | 20.00 | 10 | <**0.05**\* |
| | Correctly defined citizen science | 11.41 | 8 | 0.18 |
| | Have used citizen science in their classrooms | 2.65 | 4 | 0.62 |

\*statistically significant test result ($P \leq 0.05$).

secondary public schools were better able to correctly define citizen science than teachers from other types of school; and teachers who had heard of citizen science before were also more likely to choose the right definition (70%).

School location, subject taught and previous (personal) experience with citizen science appeared to be strong determinants of the likelihood of teachers to have used citizen science in class (Tables 2 and 3, Fig 3). In terms of school location, a significant relationship was found between location (metropolitan or rural schools), and use of citizen science in the classroom, with rural schools more likely to participate in citizen science ($X^2$ = 10.46, df = 4, p = 0.03; Fig 3). Likewise, we observed a significant correlation between the subject taught by teachers and their use of citizen science in classrooms. Science teachers (K-6 and 7–10), biology teachers, and HSIE teachers were more likely to have used citizen science in their lessons and more inclined to start or continue using citizen science in the future (Tables 3 and 4). We found no relationship between teachers' use of citizen science in classrooms, school type (comparing government and non-government schools), and age level taught (primary, secondary, middle, or combined) (Table 2).

The way teachers use citizen science varied in: frequency; the type of support received; how they learned about citizen science; and the projects in which they were involved (in both their personal lives and schools). Citizen science usage was relatively novel for some teachers, 38% reported they had been using it in class for a year or less, while 35% reported using it for over three or more years (Fig 4A). The frequency of teachers' citizen science use varied; most (44%, n = 110) reported that they used it on multiple occasions, rather than just once or consistently across a whole term or year (Fig 4B). In terms of school stages, teachers reported exploration of citizen science across all education stages, though it was most frequently employed in stage 4 (years 7 and 8) and stage 5 (years 9 and 10) (Fig 4C).

**Table 3. Subject taught, teacher's use of citizen science in classrooms, and teacher's likelihood of using citizen science in the future.**

| Subject taught | Have used citizen science | | Will start using citizen science in the future | | Will continue using citizen science in the future | |
|---|---|---|---|---|---|---|
| | $X^2$ | p-value | $X^2$ | p-value | $X^2$ | p-value |
| Science K-6 | 6.87 | **<0.05**\* | 9.45 | **<0.05**\* | 10.04 | **<0.05**\* |
| Science 7–10 | 13.58 | **<0.05**\* | 10.58 | **<0.05**\* | 12.47 | **<0.05**\* |
| Chemistry | 0.83 | 0.66 | 1.63 | 0.44 | 2.07 | 0.36 |
| Physics | 3.27 | 0.19 | 3.29 | 0.19 | 2.28 | 0.32 |
| Biology | 8.80 | **<0.05**\* | 5.60 | 0.06 | 10.61 | **<0.05**\* |
| Mathematics | 0.27 | 0.87 | 0.16 | 0.92 | 0.10 | 0.95 |
| English | 1.26 | 0.53 | 1.40 | 0.50 | 0.17 | 0.92 |
| HSIE | 8.16 | **<0.05**\* | 9.19 | **<0.05**\* | 3.39 | 0.18 |
| PDHPE | 0.35 | 0.84 | 0.41 | 0.81 | 0.13 | 0.94 |
| Creative arts | 2.45 | 0.29 | 0.08 | 0.96 | 1.37 | 0.50 |
| TAS | 0.94 | 0.62 | 1.06 | 0.59 | 1.27 | 0.53 |
| Languages | 0.93 | 0.63 | 0.55 | 0.76 | 3.47 | 0.18 |

\*statistically significant test result ($P \leq 0.05$); DF = 2.

Interestingly, despite the potential for citizen science as a tool for remote science teaching during COVID-19, the pandemic did not prompt citizen science inclusion for most respondents (73%, n = 108) notwithstanding that 57% of these teachers reported participation in citizen science before the pandemic. Although COVID-19 did not prompt the use of citizen science for respondents ($p > 0.05$), 10% of teachers started using it for the first time during the pandemic, and 17% of teachers started using citizen science more.

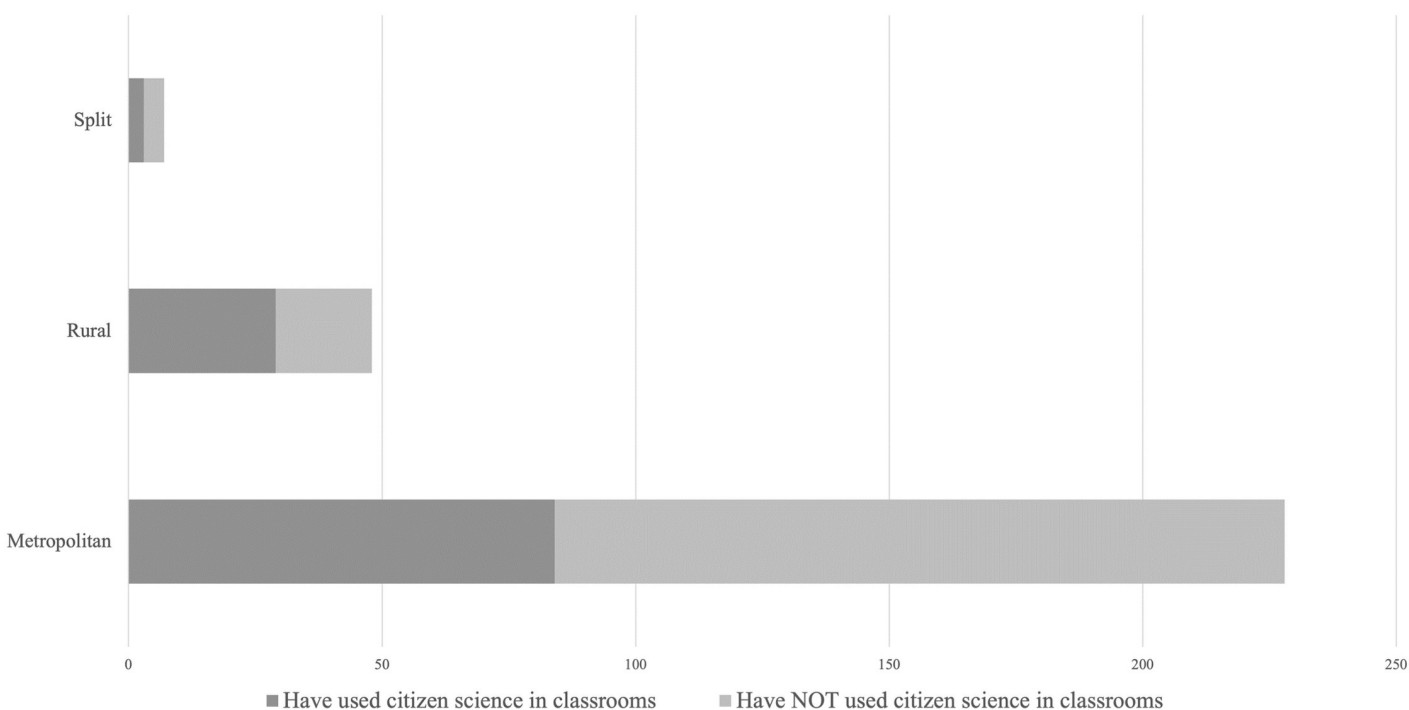

**Fig 3. School location and teacher's use of citizen science in classrooms (N = 304).**

**Table 4. Support received from research teams.**

| Identified themes | Sub-themes | Examples |
|---|---|---|
| Resources | 1) Resources from citizen science programs (kits, identification guides, teaching resources); | *"I have used resources created by* [anonymised project name] *in my classroom and online learning"*; *"Current flora and fauna data of ecosystems from council"*. |
| | 2) Resources from councils or industry (guides, brochures, lists, kits); | |
| | 3) Online resources (videos, lesson plans, information sheets); | |
| | 4) Information from Visitor Centres; | |
| | 5) Training from a citizen science project (online or in school, webinar). | |
| Curriculum mapping | Co-created citizen science project | *"I worked alongside the* [anonymised project name] *to develop a citizen science project for our year 9s"* |
| Collaboration | 1) Scientists-students' interactions (citizen science facilitators or researchers coming to schools or collaborating with the teacher); | *"We had the scientist behind the* [anonymised project name] *visit the school."*; *"Regular engagement and correspondence with other citizen scientists"* |
| | 2) Regular engagement with other citizen scientists. | |

## How teachers use citizen science

Most of the respondents who took part in citizen science in their personal lives participated in species monitoring projects in Australia, with 'Aussie Backyard Bird Count' (19%, n = 292), 'Frog ID' (14%, n = 292), 'Birds in Backyards' (11%, n = 292) and iNaturalist (10%, n = 292) cited most frequently. Similarly, the teachers who used citizen science in their lessons, participated mainly in species and weather monitoring, with 'Aussie Backyard Bird Count' (20%, n = 198), 'Frog ID' (10%, n = 198), 'Birds in Backyards' (10%, n = 198) and ClimateWatch (10%, n = 198) the four projects most commonly named.

Most teachers learned about citizen science from other teachers (20%, n = 108) and news or social media (18%, n = 108), previous experience (15%, n = 108) and the Department of Education (10%, n = 108). How teachers used citizen science also varied based on the support they received from research teams. This was expressed in terms of resources, curriculum alignment and collaboration (Table 4). Of the 26 teachers who responded to the open-ended question about the support received from citizen science project teams, 65% reported receiving resources from citizen science researchers, including guides, kits and training (online or in person); 31% reported engaging with researchers through school visits and one respondent reported the co-creation of a citizen science project designed for Year 9.

## Barriers to citizen science implementation in schools and broader use

Respondents who had not used citizen science in their personal lives reported a lack of knowledge of available projects (67%, n = 161) as the main reason for not participating, followed by lack of time, resources, and interest (Fig 5).

The most commonly reported barriers to school citizen science implementation stemmed from teachers' lack of awareness of citizen science and/or of the projects available (31%, n = 166). Lack of curriculum alignment and time were also common issues reported by respondents. Interestingly, only a small proportion of teachers (2%) identified lack of school support and resources (e.g. lack of technology) as the main barriers to citizen science use in classrooms (Fig 6). Responses to other closed-ended questions related to school support revealed that while 54% of teachers were unsure if their school would support citizen science use, 42% were confident of school support. Only 4% of teachers reported that their school would not support the use of citizen science. Additionally, one teacher mentioned that

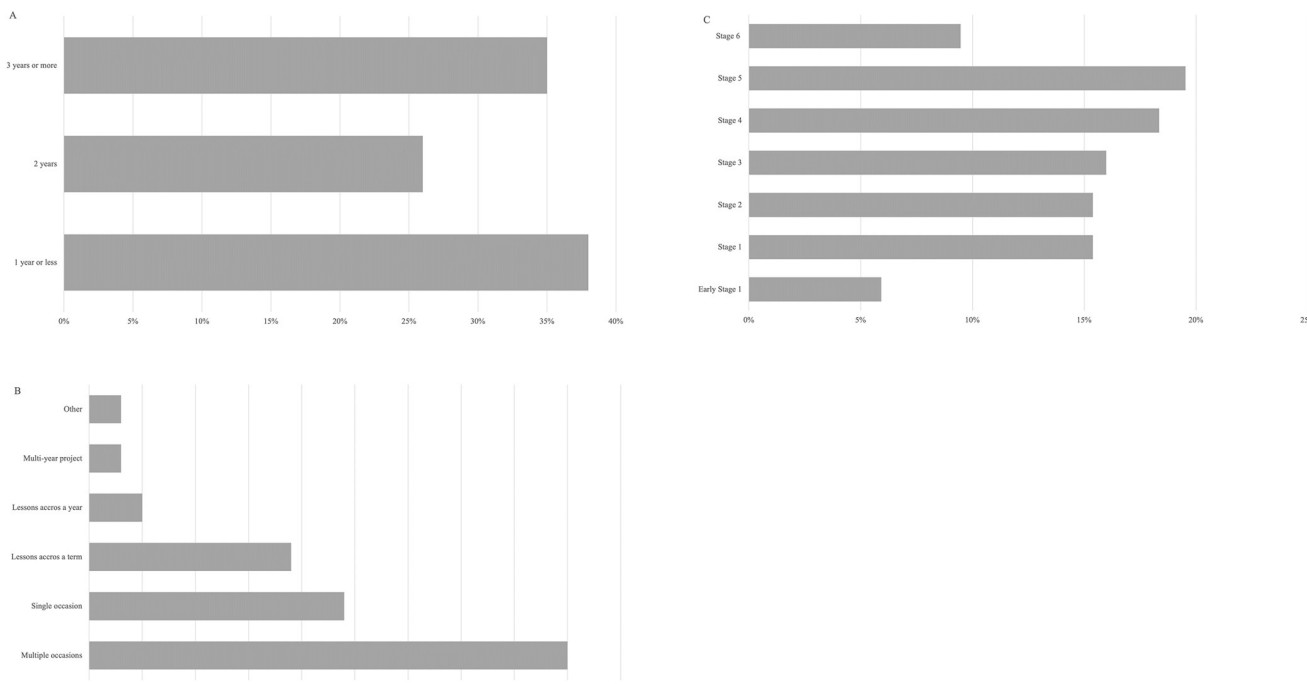

**Fig 4. Description of citizen science use in schools as reported by teachers in terms of A) Period of time, B) frequency, and C) school stage (n = 169).**

challenges imposed by lockdowns during COVID-19 reduced their ability to engage in citizen science.

## Incentives to support teachers' future participation in citizen science

Questions exploring incentives for future participation in citizen science revealed that 92% of teachers would be more likely to use citizen science programs if they had strong links to the curriculum, and that 71% would be more likely to use it if programs included interactions with research teams (n = 263) (Fig 7).

The thematic analysis revealed similarities across the 208 teachers who responded to an open-ended question regarding factors that would encourage their future participation in citizen science. Three main themes were identified by the two coders [LBS and YG], comprising 1) accessibility—increased promotion of available projects, resource availability, training and clear links of citizen science application to real life 2) curriculum mapping—the importance of links to curriculum, including those extending beyond STEM and 3) collaboration—reported needs, such as regular communication with research teams and receiving feedback (Table 5).

Teachers shared additional comments in response to the final survey question '*Is there anything else you would like to share about your involvement with citizen science*?'. Responses varied from teachers sharing their positive and valuable experiences with citizen science to those who shared that as a new term and concept, they required more information on citizen science. Two teachers commented they would specifically like to see projects linked with Indigenous Knowledge. Teachers also mentioned that citizen science projects should be interdisciplinary and applicable to local contexts.

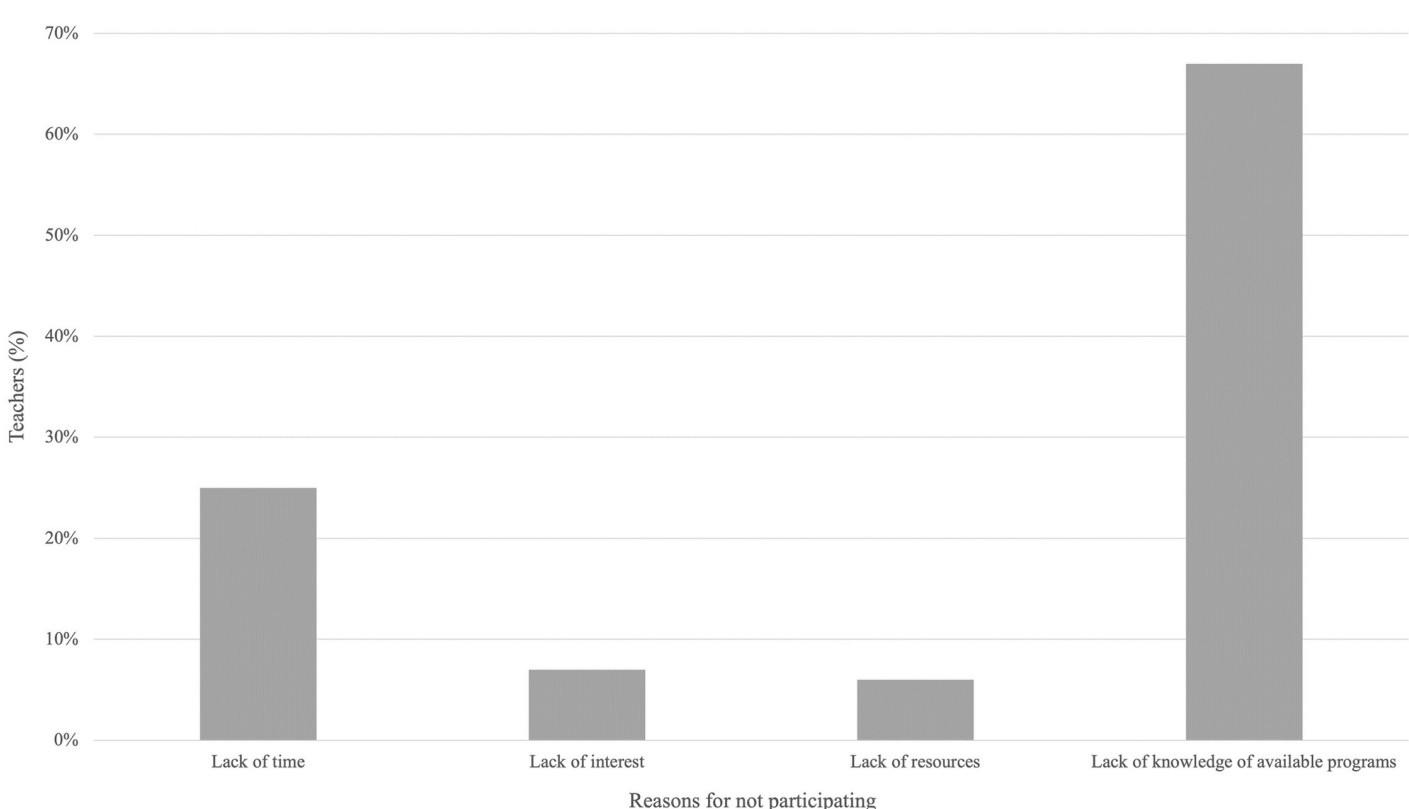

**Fig 5. Reasons for not participating in citizen science.** Teachers could select more than one option.

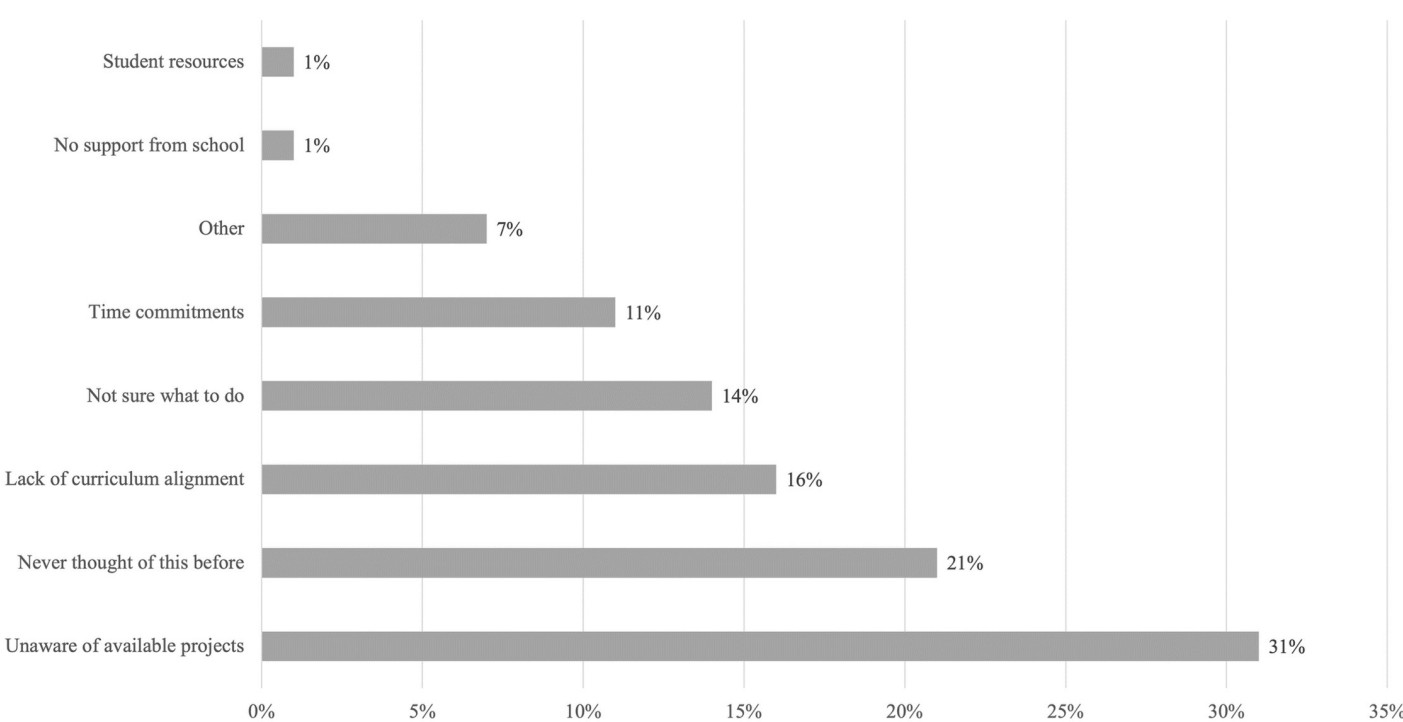

**Fig 6. Barriers to citizen science access for teachers who have not used citizen science in their lessons (n = 167).**

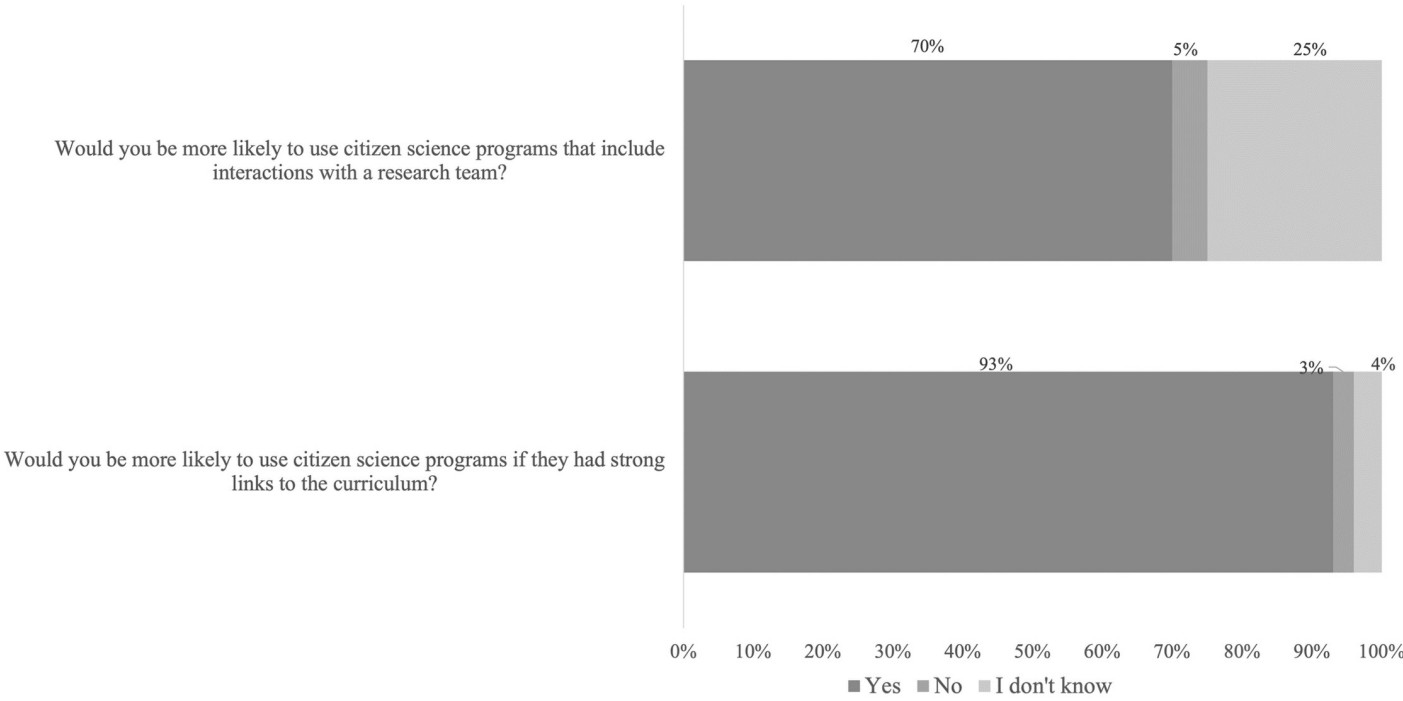

**Fig 7. Factors that may influence school use of citizen science by teachers.** *N* = 248.

## Discussion

### Teachers' knowledge and experience about citizen science

The results of this study describe the experiences, knowledge, and barriers to using citizen science in classrooms reported by teachers from across Australia. Although recent research has

**Table 5. Identified themes from the open-ended question: What would encourage and support teachers' future participation in citizen science (N = 208).**

| Themes | Sub-themes | Examples |
|---|---|---|
| Accessibility | 1) More promotion, information, knowledge; | *"Further awareness about what's available."*;<br>*"More PD and unit/lesson plans for specific projects".* |
| | 2) Clarity and applicability; | |
| | 3) Project ideas; | |
| | 4) Time; | |
| | 5) Specific training; | |
| | 6) Overall resources availability; | |
| | 7) Support from schools and institutions; | |
| | 8) Support to Indigenous students; | |
| | 9) Ease and flexibility; | |
| | 10) Safety and Privacy. | |
| Curriculum mapping | 1) Links to the curriculum; | *"Curriculum relevant science topics for the age group.*<br>*Connections to our local environment."* |
| | 2) Links to the curriculum beyond STEM. | |
| Collaboration | 1) Connectivity; | *"Collaboration and support from research institutes as well as platforms for data collection that suit indigenous* |
| | 2) Feedback; | *students for whom English is a 3rd or 4th language. i.e., heavily picture-based apps on tablets for use in the field."* |

explored teacher knowledge of citizen science and barriers to its implementation in schools [46,55,67,68], to the best of our knowledge, this study is the first to examine the current state of citizen science in Australian schools and identify the challenges and opportunities associated with its implementation. Australia has a growing tradition in citizen science [69,70], with increased participation in crowdsourcing platforms, such as iNaturalist [71] and reported benefits for the community involved, such as increased scientific literacy and behaviour changes [17]. Understanding teacher experiences and barriers to school-implementation of citizen science in Australia could help us understand similar opportunities and challenges worldwide, where citizen science is also emerging and expanding.

Around half of the teachers surveyed reported previous experience with citizen science. Also, as anticipated, teachers with responsibility in delivering science content were the most likely to include citizen science in their classrooms. This trend reflects the current state of citizen science in Australia, where the majority of the 600 plus projects listed in the Australian Citizen Science Association and Atlas of Living Australia are related to the environment and species monitoring [72,73]. However, citizen science projects with a focus extending beyond STEM disciplines are emerging worldwide and could offer an opportunity for arts and humanities teachers to incorporate this approach into their lessons. Examples of such programs involve translation of historical documents, correcting and transcribing texts or manuscripts, digitising images for digital curation [74,75], and documenting and georeferencing graveyards to understand local history [76].

In exploring the experience of teachers in citizen science through a nationwide survey, we discovered more about their knowledge and practice in such programs. Our findings showed that science teachers (K-6 and 7–10), biology teachers, and HSIE teachers had used this approach in their lessons more than other subject teachers. Citizen science within STEM not only amplifies the scope of scientific inquiry but also cultivates a hands-on, experiential learning environment for students [77,78]. The active engagement of students in citizen science projects serves as a catalyst for fostering a deeper and more nuanced understanding of scientific principles, providing a bridge between theoretical concepts and practical application [79]. In addition, promoting the benefits of citizen science in the humanities could encourage the implementation of interdisciplinary projects in schools and reach a broader range of students. Beyond supporting teaching science content, citizen science can foster the development of critical thinking and inquiry skills relevant to the entire community, including those who do not pursue scientific careers [80].

Additionally, for the purpose of our study, we employed an expert review process, where NSW Department of Education teachers or former teachers, based at Taronga Zoo, validated the content of our research instrument. To enhance the study's replicability and international relevance, we recommend that other researchers undertake a similar validation process by engaging a cohort of teachers to review the instrument and propose context-specific adaptations (e.g., subjects taught, etc.).

### The breadth and depth of citizen science in schools

As chi-square tests revealed, both government and non-government schools have similar levels of access to citizen science, and this aspect is not likely to hinder participation. Additionally, rural schools tended to participate in citizen science more frequently. This is consistent with findings that citizen science volunteers tended to be from more rural areas in comparison with more urban areas [81]. Indeed, a recent study showed that the number and diversity of citizen science projects targeting metropolitan areas is relatively low in Australia [82]. While rural schools may provide easier access to natural environments due to proximity and familiarity

with local issues, the availability of urban projects could support broader participation of metropolitan schools in citizen science.

Teachers' preferences for species and environmental monitoring initiatives may be attributed to their accessibility and ease of participation. Historically, these types of citizen science projects have been among the earliest and most widely available initiatives [14]. Nonetheless, the flexibility in observation methods reduces teachers' and schools' constraints, such as time, safety, and accessibility to resources [83,84]. Such projects are typically free or low-cost and only require basic equipment, such as notebooks, or computers or tablets, for capturing and uploading local data to platforms like eBird and iNaturalist [23,45]. We note that, where possible, it is often preferable if students collect and immediately input information using smartphones or tablets through project-specific apps or websites.

The higher incidence in junior high school and primary school can be accounted for by the flexibility of the curriculum [85]. Lower incidence of citizen science use in stage 6 could be linked to the rigour of the curriculum at this level, where students have a large amount of content to learn for final exams [85]. Low frequency in early stage 1 could be related to the lack of citizen science projects that address students' skill levels and knowledge at this age [44].

## Barriers to use of citizen science in schools

The lack of awareness of citizen science and how to use it, were found to be the main barriers to its implementation, aspects that could be overcome by an increase in school-based citizen science projects piloted and professional development training with teachers, as discussed in recent research [55]. There is also a need to increase teachers' awareness of the benefits of citizen science for students. In alignment with previous research [55,86] the present study [56] also reported the lack of time and curriculum alignment as the main challenges in implementing citizen science in schools. Teachers may be more likely to incorporate citizen science into the classroom if they link citizen science purpose and design to the Australian curriculum as well as develop meaningful, contextual, and practical projects.

Non-science teachers reported a lack of curriculum alignment as the main barrier to citizen science implementation much more frequently than science teachers, which is to be expected as most citizen science projects can, with some work and creativity, be aligned to the science curriculum already [87,88]. As previously reported by educators and leaders in citizen science, a challenge lies in the inherent difficulty of aligning projects with students due to their original design not being expressly intended for educational purposes [56]. This challenge underscores the need for a deliberate and purposeful alignment strategy to enhance the educational utility of citizen science initiatives in school settings. Co-creating projects that feature interdisciplinary synergies between social sciences, humanities, and STEM could support the emergence of more diverse and inclusive citizen science school projects [38].

Additionally, this study identified a lack of curriculum alignment as the main barrier to citizen science implementation in classrooms. Developing a framework that explicitly maps citizen science projects to educational standards is crucial for overcoming this obstacle and facilitating effective classroom integration. Previous research has developed resources to align citizen science projects with the Australian curriculum, aiding teachers in classroom implementation [87]. Various citizen science projects were mapped to the NSW Stage 4 syllabus themes (Physical World, Earth and Space, Living World, and Chemical World) while also highlighting their contributions to skill development and curriculum integration. This guide illustrates how the Big City Birds project, for instance, aligns with the Australian curriculum outcome *SC414LW*: *relating the structure and function of living things to their classification, survival, and reproduction* [88]. Developed by [YG], [CP], and [AM], this resource represents the

initial phase of the Learning By Doing initiative. Future steps will include: 1) detailed curriculum mapping, 2) expanding to additional educational stages, and 3) aligning with syllabi in other Australian states and territories.

## Incentives to support teachers' future participation in citizen science

Although the vast majority of respondents were likely to use citizen science in the future, a quarter of those surveyed expressed uncertainty about whether they would be inclined to do so. It is possible that this is the result of insufficient support provided by researchers to teachers. Only 28% of teachers who have participated in citizen science have used resources or received support from a research team. This corroborates previous research, as support from researchers to help teachers implement activities was mentioned in focus groups to be very important [56].

Critically, teachers reported that they wanted a more clear understanding on how the data being contributed to citizen science projects apply to real life. Teachers shared that they would like to have specific training in citizen science with school and research institution support. These results link to the main barriers to using citizen science in schools, especially to the lack of knowledge of available programs. Thus, engaging the current and next generations of teachers with regard to the benefits of school-based citizen science could expand its implementation in Australia. As such, we recommend a Unit of Study in citizen science designed and delivered for undergraduate education students (teachers in training) to support the long-term sustainability of school-based citizen science in Australia. Although citizen science projects are starting to be integrated into teacher professional development and preparation courses (see 68, 90), such initiatives are still relatively scarce in the literature. In 2022, five universities in Spain engaged over 350 pre-service Early Childhood and Primary Teachers in a soil health citizen science program [89], while a university in South Australia provided professional learning in species monitoring to middle school teachers [34]. Additional local initiatives worldwide have demonstrated the benefits of such programs [68,90], underscoring the need for broader implementation of professional development opportunities in citizen science for teachers in Australia and globally [29,91].

Finally, formal curriculum links have been shown to increase the benefits of citizen science projects [35]. A study in Spain has shown that citizen science integrated into the curriculum improved students' interest, curiosity, and appreciation for the sciences, supporting learning and enabling relatability to real life [92]. Additionally, curriculum alignment may help with challenges presented by citizen science project leaders [56].

## Research limitations

The teacher scoping questionnaire had a number of limitations, particularly associated with the sampling method. The ethics application was approved when a large part of the country was forced into lockdowns due to the COVID-19 pandemic, in 2020, and schools were teaching remotely online. At first the team delayed questionnaire distribution, as the school lockdowns were thought to be a short-term measure. However, when lockdowns were extended, the questionnaire was distributed to minimise delays to the research. At this time teachers were even busier than usual while teaching their students remotely, and it is thought this may have limited the sample size. The snowball sampling technique may have introduced selection bias into the sample, as many participants are contacts of researchers who may not represent a broad sample of teachers nationally [62]. Similarly, it is thought that the sampling method resulted in a skewed response level from New South Wales, with the overall dataset less representative of the entirety of Australia. This skew to NSW was, however, useful for the project

team's implementation of citizen science programs within schools in the state. The responses were also biased in terms of demographics. The demographics of respondents corroborate the high proportion of women teachers in Australia [93]. Future studies could seek a larger number of responses from every state and territory in Australia.

## Conclusion

Our results show that linking citizen science purpose and design to the Australian curriculum, and building meaningful, contextual and applicable projects could contribute to teachers being more likely to include citizen science in their classrooms. Of the teachers who participated in the questionnaire, 40% use or have used citizen science in their classes. This number is thought to be higher than actual usage rates across Australia due to questionnaire sampling, with 52% of the teachers surveyed sharing that they knew what was meant by citizen science. Prior personal involvement predicted classroom use of citizen science. Rural schools participated to a greater extent than metropolitan schools. And science teachers have used citizen science significantly more than non-science teachers.

Teachers' use of citizen science is varied. It is used across all curriculum stages, with highest use in stage 1 through to stage 5. Early stage 1 and stage 6 teachers used citizen science to a lesser extent, possibly due to the demands associated with low levels of skill development and curriculum constraints respectively. Teachers' use of citizen science tended to be short term, either across multiple occasions or a single use, with only a small proportion of teachers using it for longer periods. Additionally, citizen science use was relatively new for many teachers, with 38% using it for a year or less, however there were some patterns of long-term use with 35% of teachers having used it for 3 or more years. Teacher-use patterns can inform school citizen science project planning.

Teachers reported a variety of barriers to citizen science participation. The most frequent barriers (a lack of awareness, never having thought of it, and lack of curriculum alignment) indicate two major issues for citizen science uptake in schools. Teachers are unaware of the opportunities that are out there and how to access them, while current projects are not often designed to suit teachers and fit with curriculum requirements. There was no significant relationship between the main barriers and location or school type, but non-science teachers selected curriculum alignment as a barrier more than anticipated by our research team. Additionally, teachers strongly agreed that greater curriculum links and support from a research team would benefit their participation in citizen science. These barriers indicate there needs to be more communication about citizen science with teachers, but also that projects could be aligning to curriculum elements beyond science to increase involvement and engagement.

As citizen science can contribute to STEM career motivation, it is important to understand how likely both experienced and recently-qualified teachers are to explore this approach in their classrooms. Our research findings contribute to a broader understanding of how citizen science has been used in Australian schools and highlight the challenges and opportunities for its implementation. Moreover, insights from our research provide valuable contributions to the international discourse on citizen science in educational settings, suggesting that the challenges and opportunities identified in Australian schools may offer instructive parallels for global contexts. Finally, as part of the LBD project, these results help us refine the next steps for citizen science implementation in schools.

In conclusion, our findings underscore the need for citizen science projects to align more closely with the specific needs of schools, emphasising the necessity for responsiveness in project design and implementation. To address this, the LBD project has proactively undertaken measures to mitigate existing barriers. These efforts involve demonstrating a commitment to

upscaling the availability of citizen science projects within educational contexts and building more effective partnerships with schools. As we advocate for alignment between projects and educational needs, it is essential to explore collaborative models where both researchers and teachers contribute their expertise. This emphasises the need for a shared responsibility, acknowledging that bridging the gap between scientific research and educational requirements necessitates a concerted effort from both communities. As our previous research has highlighted, researchers often lack educational insights and an understanding of the school context, and teachers may face constraints including time and a lack of prior opportunity to engage in scientific research. This challenge, while significant, presents an opportunity for the scientific and educational communities to collaboratively devise innovative solutions, fostering a symbiotic relationship that ensures the meaningful integration of citizen science into school settings.

## Supporting information

**S1 File.**
(PDF)

**S2 File.**
(XLSX)

**S3 File.**
(DOCX)

## Acknowledgments

We acknowledge and pay respect to the Gadigal people of the Eora Nation, the Traditional Owners of the land on which we research, teach, and collaborate at The University of Sydney. We acknowledge all teachers' efforts in contributing to this research by completing the questionnaire. The authors acknowledge the Statistical Consulting Service provided by Alexandra Green from the Sydney Informatics Hub, a Core Research Facility of the University of Sydney. We also acknowledge teachers from the Taronga Zoo for their time piloting and validating the covered questions. We thank the thoughtful reviewers of this paper for their helpful comments.

## Author Contributions

**Conceptualization:** Ciara Kenneally, Yaela Golumbic, John M. Martin, Christine Preston, Peter Rutledge, Alice Motion.

**Data curation:** Ciara Kenneally, Yaela Golumbic, John M. Martin, Christine Preston, Peter Rutledge, Alice Motion.

**Formal analysis:** Larissa Braz Sousa, Yaela Golumbic, John M. Martin, Christine Preston, Peter Rutledge, Alice Motion.

**Funding acquisition:** Yaela Golumbic, John M. Martin, Christine Preston, Alice Motion.

**Investigation:** Ciara Kenneally, Yaela Golumbic, John M. Martin, Christine Preston, Peter Rutledge, Alice Motion.

**Methodology:** Ciara Kenneally, Yaela Golumbic, John M. Martin, Christine Preston, Peter Rutledge, Alice Motion.

**Project administration:** Yaela Golumbic, John M. Martin, Alice Motion.

**Resources:** Larissa Braz Sousa, Ciara Kenneally, Yaela Golumbic, John M. Martin, Christine Preston, Peter Rutledge, Alice Motion.

**Software:** Larissa Braz Sousa, Yaela Golumbic.

**Supervision:** Yaela Golumbic, John M. Martin, Peter Rutledge, Alice Motion.

**Validation:** Larissa Braz Sousa, Yaela Golumbic, John M. Martin, Christine Preston, Peter Rutledge, Alice Motion.

**Visualization:** Larissa Braz Sousa.

**Writing – original draft:** Larissa Braz Sousa, Ciara Kenneally, Yaela Golumbic, John M. Martin, Christine Preston, Peter Rutledge, Alice Motion.

**Writing – review & editing:** Larissa Braz Sousa, Yaela Golumbic, John M. Martin, Christine Preston, Peter Rutledge, Alice Motion.

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
