## [Decision Letter · Decision Letter 0]

24 Jul 2024

PONE-D-23-42215Teacher experiences and understanding of citizen science in Australian classroomsPLOS ONE

Dear Dr. Braz Sousa,

Thank you for submitting your manuscript to PLOS ONE. After careful consideration, we feel that it has merit but does not fully meet PLOS ONE’s publication criteria as it currently stands. Therefore, we invite you to submit a revised version of the manuscript that addresses the points raised during the review process.

We look forward to receiving your revised manuscript.

Kind regards,

José Gutiérrez-Pérez

Academic Editor

PLOS ONE

Journal Requirements:

5. We note that you have indicated that there are restrictions to data sharing for this study. For studies involving human research participant data or other sensitive data, we encourage authors to share de-identified or anonymized data. However, when data cannot be publicly shared for ethical reasons, we allow authors to make their data sets available upon request. For information on unacceptable data access restrictions, please see http://journals.plos.org/plosone/s/data-availability#loc-unacceptable-data-access-restrictions. 

Reviewers' comments:

Reviewer's Responses to Questions

**Comments to the Author**

1. Is the manuscript technically sound, and do the data support the conclusions?

Reviewer #1: Yes

Reviewer #2: Yes

2. Has the statistical analysis been performed appropriately and rigorously? 

Reviewer #1: Yes

Reviewer #2: Yes

3. Have the authors made all data underlying the findings in their manuscript fully available?

Reviewer #1: No

Reviewer #2: No

4. Is the manuscript presented in an intelligible fashion and written in standard English?

Reviewer #1: Yes

Reviewer #2: Yes

5. Review Comments to the Author

Reviewer #1: The manuscript explores teacher experiences and their understanding of citizen science in Australian classrooms using mixed-methods approach of questionnaires with descriptive data. Through the data collected, the authors analyse teachers’ past experience of citizen science, perceived barriers in implementing such projects in school, and their views on supportive policies or resources to increase participation of teachers in citizen science. The findings point to some interesting insights such as rural schools being more engaged in citizen science, possibly due to proximity to nature as discussed by the authors. However, there may be other factors that might need additional probing. Lack of awareness regarding citizen science projects and absence of clear alignment with curricula were cited as barriers to implementation by a majority of teachers, as compared to lack of infrastructure.

Overall, the manuscript provides a fairly descriptive account of the contours citizen science projects in schools through teachers, but as a reader, there is an expectation for more in-depth analysis regarding the issues mentioned. For instance, the authors mention bird counting projects as being most popular. What features of this activity allows better engagement at teacher and student level? Are citizen science projects a part of teacher professional development or preparation courses? What assumptions do the authors hold when they question teachers regarding their knowledge of citizen science? Similarly, are there any citizen science projects developed with an explicit focus on pedagogy? The descriptive data presented isn’t entirely unexpected, and the questions raised call for some suggestions regarding curricular mapping. It would be extremely useful if the authors could provide a rudimentary framework for such mapping using some illustrative examples from middle-school science subjects based on the responses received.

I understand that these are broad suggestions, and it may not be possible to address them all. However, the main aim here to push the authors to go beyond the descriptive data and expand on possible discussions. I would argue that the real value-addition of the manuscript lies in outlining the possibilities that could form a part of future policies and teacher preparation courses.

Reviewer #2: The manuscript addresses a topic of high contemporary interest. The literature review is current and appropriate to the topic. The methodological approach, from a mixed approach, allows us to answer several questions: investigated four focal research questions: 1) What knowledge and experience do teachers have of citizen science? 2) How have

teachers used citizen science in their personal lives and with their students? 3) What are the barriers to citizen science implementation in schools and its broader use? and 4) What would encourage and support the future participation of teachers in citizen science?

It is suggested that the authors attend in more detail to the quality criteria of the instrument used. Mention is made of the content validity process of the instrument, but no relevant information about the reliability of the instrument is appreciated.

We suggest to the authors a complementary analysis of the grouping of the items (e.g. exploratory factor analysis) that would provide relevant information for potential replications of the study, increasing its international interest.

6. PLOS authors have the option to publish the peer review history of their article (what does this mean?). If published, this will include your full peer review and any attached files.

Reviewer #1: **Yes: **Deborah Dutta

Reviewer #2: No

---

## [Author Response · Author response to Decision Letter 0]

20 Sep 2024

Response to reviewers’ comments

Reviewer #1:

1. For instance, the authors mention bird counting projects as being most popular. What features of this activity allows better engagement at teacher and student level?

Thank you for your valuable comment. This has been addressed on lines 435-443. 

Teachers’ preferences for species and environmental monitoring initiatives may be attributed to their accessibility and ease of participation. Historically, these types of citizen science projects have been among the earliest and most widely available initiatives (Miller-Rushing et al., 2012). Nonetheless, the flexibility in observation methods reduces teachers’ and schools’ constraints, such as time, safety, and accessibility to resources (Silvertown, 2009, Wood et al., 2011). Such projects are typically free or low-cost and require only basic equipment, such as notebooks, or computers or tablets, for capturing and uploading local data to platforms like eBird and iNaturalist (Soanes et al., 2020, Dickinson et al., 2012). We note that, where possible, it is often preferable if students collect and immediately input information using smartphones or tablets through project-specific apps or websites.

2. Are citizen science projects a part of teacher professional development or preparation courses? 

This question has been addressed on lines 491-506.

Critically, teachers reported that they wanted a more clear understanding on how the data being contributed to citizen science projects apply to real life. Teachers shared that they would like to have specific training in citizen science with school and research institution support. These results link to the main barriers to using citizen science in schools, especially to the lack of knowledge of available programs. Thus, engaging the current and next generations of teachers with regard to the benefits of school-based citizen science could expand its implementation in Australia. As such, we recommend a Unit of Study in citizen science designed and delivered for undergraduate education students (teachers in training) to support the long-term sustainability of school-based citizen science in Australia. Although citizen science projects are starting to be integrated into teacher professional development and preparation courses, such initiatives are still relatively scarce in the literature. In 2022, five universities in Spain engaged over 350 pre-service Early Childhood and Primary Teachers in a soil health citizen science program (Eugenio-Gozalbo et al., 2022), while a university in South Australia provided professional learning in species monitoring to middle school teachers (Paige et al., 2015). Additional local initiatives worldwide have demonstrated the benefits of such programs (Scheuch et al., 2018, Huffling and Scott, 2021), underscoring the need for broader implementation of professional development opportunities in citizen science for teachers in Australia and globally (Roche et al., 2020, Harlin et al., 2018).

3. What assumptions do the authors hold when they question teachers regarding their knowledge of citizen science? 

This question has been addressed on lines 166-170.

When questioning teachers regarding their knowledge of citizen science, the authors assume teachers to have varying levels of familiarity with the concept and practice of citizen science and that teachers’ understanding of citizen science may influence their willingness and ability to incorporate such projects into their teaching practices.

4. Similarly, are there any citizen science projects developed with an explicit focus on pedagogy?

Thank you for your question. It has been added to the lines 74-79.

There are, however, limited examples of f citizen science projects that are designed to integrate pedagogical objectives and approaches, aiming to enhance teaching practices and learning outcomes within educational settings (Harlin et al., 2018, Bopardikar et al., 2023). Examples include programs that provide structured teaching resources and adaptable lessons to support schools’ pedagogical practices and curricula (Kocman et al., 2020, Bopardikar et al., 2023). Such programs have been especially reported in the fields of environmental science and biodiversity (Spicer et al., 2020, Aivelo, 2023). 

5. The descriptive data presented isn’t entirely unexpected, and the questions raised call for some suggestions regarding curricular mapping. It would be extremely useful if the authors could provide a rudimentary framework for such mapping using some illustrative examples from middle-school science subjects based on the responses received.

Thank you for the suggestion. The initial framework developed by our team has been added to lines 469-482.

Additionally, this study identified a lack of curriculum alignment as the main barrier to citizen science implementation in classrooms. Developing a framework that explicitly maps citizen science projects to educational standards is crucial for overcoming this obstacle and facilitating effective classroom integration. Previous research has developed resources to align citizen science projects with the Australian curriculum, aiding teachers in classroom implementation (Learning By Doing, 2021b). Various citizen science projects were mapped to the NSW Stage 4 syllabus themes (Physical World, Earth and Space, Living World, and Chemical World) while also highlighting their contributions to skill development and curriculum integration. This guide illustrates how the Big City Birds project, for instance, aligns with the Australian curriculum outcome SC414LW: relating the structure and function of living things to their classification, survival, and reproduction (Learning By Doing, 2021a). Developed by [YG], [CP], and [AM], this resource represents the initial phase of the Learning By Doing initiative. Future steps will include: 1) detailed curriculum mapping, 2) expanding to additional educational stages, and 3) aligning with syllabi in other Australian states and territories.

Reviewer #2:

1. It is suggested that the authors attend in more detail to the quality criteria of the instrument used. Mention is made of the content validity process of the instrument, but no relevant information about the reliability of the instrument is appreciated. We suggest to the authors a complementary analysis of the grouping of the items (e.g. exploratory factor analysis) that would provide relevant information for potential replications of the study, increasing its international interest.

Thank you for your suggestion. While exploratory factor analysis (EFA) can be a valuable tool for identifying underlying structures within a dataset, it may not be suitable for our data. Our study is focused on categorical variables, and EFA is typically more appropriate for continuous data where underlying factors are being explored. Therefore, applying EFA in this context may not yield meaningful results. To enhance the study’s replicability and international relevance, other approaches more aligned with the nature of our data would be advisable. For the context of our study, we used Expert Review with Taronga Zoo teachers to validate the content of our research instrument. We recommend that researchers in other countries validate the questionnaire by inviting a small cohort of local or national teachers to review the instrument and suggest adaptations based on local contexts. This has been added to our discussion on lines 418-423.

---

## [Editor Report · Decision Letter 1]

11 Oct 2024

Teacher experiences and understanding of citizen science in Australian classrooms

PONE-D-23-42215R1

Dear Dr. Braz,

We’re pleased to inform you that your manuscript has been judged scientifically suitable for publication and will be formally accepted for publication once it meets all outstanding technical requirements.

Kind regards,

José Gutiérrez-Pérez

Academic Editor

PLOS ONE

---

## [Editor Report · Acceptance letter]

31 Oct 2024

PONE-D-23-42215R1 

PLOS ONE

Dear Dr. Braz Sousa, 

I'm pleased to inform you that your manuscript has been deemed suitable for publication in PLOS ONE. Congratulations! Your manuscript is now being handed over to our production team.

Kind regards, 

on behalf of

Dr. José Gutiérrez-Pérez 

Academic Editor

PLOS ONE